# Efficiency of Thyme and Oregano Essential Oils in Counteracting the Hazardous Effects of Malathion in Rats

**DOI:** 10.3390/ani14172497

**Published:** 2024-08-28

**Authors:** Fatimah A. Al-Saeed, Sayed Soliman Abd-Elghfar, Montaser Elsayed Ali

**Affiliations:** 1Department of Biology, College of Science, King Khalid University, Abha 61413, Saudi Arabia; falsaed@kku.edu.sa; 2Department of Animal Productions, Faculty of Agriculture, Al-Azhar University, Cairo 11651, Egypt; sayed.soliman@azhar.edu.eg; 3Department of Animal Productions, Faculty of Agriculture, Al-Azhar University, Assiut 71524, Egypt

**Keywords:** hazards, malathion, oregano essential oils, thyme essential oils, toxicity

## Abstract

**Simple Summary:**

Malathion (MLT) is one of the most widely used pesticides and may pose multiple hazards to humans and animals. Some therapeutic herbs have the potential to protect against these toxic activities. In this paper, we looked at thyme and oregano essential oils (ThEO and OEO, respectively) as possible compounds for anti-toxic therapies following MLT exposure. This paper offers an experimental therapeutic approach using natural antitoxins, thereby providing a solution to the problems of raising livestock that are exposed to nutritional toxicity.

**Abstract:**

The widespread use of MLT may pose numerous hazards to animal breeding, health, and resilience due to the presence of MLT residues in animal feedstuffs, pastures, hay, and cereals. Many medicinal plants provide what is called a generalized anti-toxic remedy. The current study examined hazardous biochemical and histological reactions to MLT and the efficiency of ThEO and OEO essential oils as anti-toxic therapies to return to a natural state after MLT exposure. A total of 75 male albino rats were randomly assigned to two groups: (i) C − MLT, comprising 25 rats, served as the control group; and (ii) C + MLT, with 50 rats that were exposed to 5 mg/kg/BW. After exposure to MLT for 21 days, a return to normal status was determined by subdividing the C + MLT group into two equal groups: ThEO and OEO were used as treatments, with 100 mg/kg body weight of thyme and oregano essential oils, respectively, being administered for 21 days. The results showed a significant decrease in body weight gain (BWG) and final weight (FW) compared to C − MLT, while the therapeutic effects of ThEO and OEO enhanced FW and BWG. Our results indicated that MLT exposure resulted in deficient serum liver function, but that OEO and ThEO therapy brought about a significant improvement in liver enzyme function. Although there was no significant difference in serum aspartate transaminase (AST) or alkaline phosphatase (ALK-Ph) and a significant drop in alanine transaminase (ALT) and acetyl choline-esterase (AChE) levels, the C + MLT group showed hepatic fibrosis in the third stage. Furthermore, histological sections of the OEO and ThEO groups showed reduced hepatocellular damage, inflammation, and hepatic fibrosis. However, there was a significant increase in serum creatinine between the C + MLT and C − MLT groups following exposure to MLT. Histological sections of renal tissue from rats treated with OEO and ThEO showed reduced tubular damage, reduced interstitial inflammation, and preserved renal tissue architecture. In conclusion, OEO and ThEO are potential compounds for use as anti-toxic therapies to return to a natural state after MLT exposure. These compounds could serve as an experimental therapeutic approach against natural toxins, providing a solution to the problems of raising livestock that are exposed to nutritional toxicity.

## 1. Introduction

Malathion (MLT), an organophosphorus insecticide, is one of the most widely used insecticides [1]. The most widespread chemical form of MLT is an amber-colored liquid, designated diethyl 2-[(dimethoxyphosphorothioyl) sulfonyl] butanedioate [2]. The toxicity of MLT toxicity is due to its mode of action as an acetylcholinesterase (AChE) inhibitor. The inhibition of AChE causes acetylcholine to accumulate at muscarinic and nicotinic sites, resulting in acute hyperstimulation due to the persistent presence of the neurotransmitter [3].

MLT has been utilized to control livestock parasites, insects on pets, gardens, and farm crops, mosquitoes, and Mediterranean fruit flies and as an ultra-low volume spray in urban areas [4,5]. MLT residues may be transmitted to animals through feedstuffs, pastures, hay, cereals, and other feed sources, posing risks to animal welfare, health, and breeding [6]. There is a high risk of MLT toxicity in humans because very small amounts are required to cause toxic effects [7]. The situation is intensified by the fact that exposure occurs when it is used continuously over a long period of time or when it is applied over large areas during crop production and in post-harvest commodities [8].

Among therapeutic herbs, some have the ability to protect organisms against toxic actions [9], i.e., so-called generalized anti-toxic remedies [10]. Recent studies have shown that oregano and thyme essential oils (OEO and ThEO) contain natural antioxidants [11]. The administration of specific natural antioxidants may be an effective method of protecting cellular membranes and biomolecules against oxidative and/or toxic stress-related damage in a biological system [12]. In an in vitro study, a thyme/carvacrol mixture was shown to inhibit the production of patulin mycotoxin [13]. The compounds from oregano oils have been shown to display several biological properties that might be useful in managing aflatoxin-associated health problems [14]. The properties of OEO and ThEO may help protect against toxins and fight infections [15,16].

Misuse by rural farmers has been linked to MLT toxicity risks in humans and animals. However, in developed countries, there is a high incidence of home users being impacted by eating food treated with MLT-containing insecticides [7]. Also, animals that consume MLT-contaminated feed and grains may also pass it on through their products [6]. This study was carried out to investigate hazardous biochemical and histological reactions to MLT and the efficiency of ThEO and OEO essential oils as anti-toxic therapies to return to a natural state after MLT exposure. Such knowledge may be a step toward developing potentially unique treatment options using natural antitoxins.

## 2. Materials and Methods

### 2.1. Animals

Seventy-five male albino rats, healthy and clinically free of diseases, with a body weight of 134.95 ± 0.59 g and aged 50 d, were included in this study. Rats were obtained from El Osman Farm, Cairo, Egypt. The rats were kept in stainless steel cages and housed under standard conditions of temperature (23 °C) and lighting (12 h; light/dark cycles), with free access to food and drinking water ad libitum.

### 2.2. Exposure to MLT Toxicity

Rats were randomly assigned to two groups: (i) C − MLT, comprising 25 rats, served as the control group; and (ii) C + MLT, comprising 50 rats which were dosed at 5 mg/kg body weight MLT (volume: 1 mL; MLT LD50: 1/316) for 21 days. Malathion-d10 (diethyl D10, 100 µg/mL^−1^ in acetonitrile, 99.0% isotopic purity; product code: DLM-4476-S; Cambridge, MA, USA). MLT was obtained from the Kafr El-Zayat Company for Chemicals and Pesticides in Kafr El-Zayat, Egypt. The compound is soluble in water and in most organic solvents.

### 2.3. Return to Normal Status after Exposure to MLT Toxicity

C − MLT was continued from the earlier experiment, while MLT exposure was stopped after 21 days. The return to a normal state was determined by dividing C + MLT into two equal subgroups (OEO and ThEO) and treating each with 100 mg/kg body weight for 21 days. Thyme and oregano essential oils were obtained from Monachem, a strategic partner for specialty chemicals and a certified contract manufacturer in India (Batch No. TO/CAL/5021/21-22). Certificates of ThEO and OEO analyses are provided in Table 1. A schematic diagram of the experimental protocol is shown in Figure 1.

### 2.4. Biological Evaluation

The biological evaluation of the different groups was carried out by determining the initial (IW) and final (FW) body weights, as well as determining body weight gain (BWG) using the following formulas: BWG = final weight minus beginning weight.

### 2.5. Biochemical Assays

Samples of blood from each animal from each treatment group were taken using a capillary tube to drain the ocular venous plexuses 21 days after the animal had been exposed to malathion and 42 days after the final OEO and ThEO therapy. After centrifuging the blood samples for 20 min at 3000 rpm, sera and plasma samples were extracted and kept at −20 °C until additional tests were conducted. Serum total protein and albumin levels were determined using SPINREACT kits from Chemical Company, Girona, Spain [17]. Bergmeyer’s guidelines for assaying alkaline phosphatase (ALK-Ph; U/L) and acetylcholinesterase (AChE; U/L) were followed, as in Young [18]. Quantities of serum total cholesterol (TC), triglycerides (TG), high-density lipoprotein (HDL), and very low-density lipoprotein cholesterol (VLDL-c) were calculated in mg/dL [19,20]. BioSystems Kets performed the aspartate transaminase (AST) and alanine transaminase (ALT) assays according to the method described by Low et al. [21,22]. The colorimetric kinetic approach was used to test total antioxidants (TAC), butyrylcholinesterase, glutathione peroxidase (GPx), malondialdehyde (MDA), and superoxide dismutases (SODs); Bio-Diagnostic Com [23,24].

### 2.6. Histopathological Examination

After 21 and 42 days of experimentation, five animals from each group were randomly euthanized to explore potential histological alterations. Specimens from the liver and kidney were taken and stored in a neutral buffer with 10% formalin. The samples were dehydrated with increasing alcohol concentrations, cleaned in xylene, embedded in paraffin wax, sliced at 4 μm, stained with hematoxylin and eosin, and viewed under a light microscope [25]. The images were captured with the LABOMED Fluorescence Microscope L×400, cat no. 9126000, Meubon, Houston, TX, USA. The image scale bar is 100 µm. The magnification powers were ×100 and ×400.

### 2.7. Statistical Analysis

The submitted data were statistically analyzed using SPSS version 25 for Windows 25. The normal distribution of the data was validated by the Kolmogorov-Smirnov test. The independent sample *t*-test was employed as a parametric test to compare C − MLT and C + MLT. Body weight and biochemical assay data were analyzed using a one-way ANOVA, followed by Duncan’s multiple range test [26] according to the following general linear model: Yij = µ + T_i_ + A_j_ + E_ij_, where Y_ij_ = experimental observation, µ = general mean, T_i_ = effect of treatments (i = C − MLT, OEO and ThEO), and e_ij_ = experimental error. Values ≤ 0.05 were considered to be significant.

## 3. Results

### 3.1. Biological Evaluation

The initial weight (IW), final weight (FW), and body weight gain (BWG) in C − MLT and C + MLT during MLT exposure, as well as C − MLT, OEO, and ThEO during return to normal condition in albino male rats, are presented in Table 2. The results of the independent *t*-test showed a significant (*p* < 0.001) decrease in C + MLT compared to C − MLT in the FW and BWG after MLT exposure. However, this study indicated that the therapeutic effects of thyme and oregano oils increased (*p* < 0.001) the FW and (*p* < 0.009) the BWG in the OEO and ThEO groups during their return to normal status after exposure to MLT.

### 3.2. Biochemical Assays

#### 3.2.1. Liver Function Enzymes

The liver functions of albino male rats exposed to MLT and their return to normal conditions are presented in Table 3. There was a significant increase in serum liver functions (ALT, AST, ALK-ph, and AChE) between C − MLT and C + MLT (*p* < 0.001) during exposure to MLT. In this study, the Duncan test showed a significant decrease in the levels of liver function enzymes (ALT, AST, ALK-ph, and AChE) in ThEO compared to C − MLT during the return to normal status. No significant difference was found between the OEO and C − MLT groups in terms of serum AST and ALK-ph (*p* > 0.05), while there was a significant decrease in ALT and AChE (*p* < 0.001).

#### 3.2.2. The Kidney Functions Enzymes and Lipid Profile

The kidney function enzymes and lipid profiles of albino male rats exposed to MLT and their return to normal conditions are presented in Table 4. There was a significant increase in serum T. protein, Alb, Glb, and creatinine between C − MLT and C + MLT during exposure to MLT. For the lipid profile, no significant difference was found in serum TG or VLDL (*p* > 0.05) between the C − MLT and C + MLT groups, while there was a significant decrease in TC, HDL, and LDL (*p* < 0.001). However, there was a significant decrease (*p* < 0.001) in serum Alb and Glb among OEO and ThEO compared to C − MLT. Also, the lipid profile showed a significant decrease (*p* < 0.001) in serum TC, TG, HDL, and LDL for OEO and ThEO compared to C − MLT during the return to normal status after MLT exposure, while no significant difference was found in serum creatinine, T. protein, VLDL, or AI (*p* > 0.05) between the C − MLT and therapeutic groups.

#### 3.2.3. Antioxidant Profiles

The antioxidant profiles (GPX, BCA, TAC, SOD, and MDA) during exposure to malathion and return to normal status in the albino male rats are shown in Table 5. In the present study, there was a significant decrease (*p* < 0.001) in serum TAC, GPX, and SOD in the C + MLT group compared to the C − MLT group. At the same time, BCA and MDA showed a significant increase (*p* < 0.001) in the C + MLT compared to the C − MLT during exposure to MLT. Also, the antioxidant profile showed no significant difference (*p* > 0.05) in serum TAC, SOD, BCA, and MDA between ThEO and C − MLT during the return to normal status after MLT exposure. Meanwhile, the OEO therapeutic group had a significantly higher serum antioxidant profile than C − MLT.

#### 3.2.4. Complete Blood Counts (CBCs)

The complete blood counts (CBCs) during exposure to MLT and return to normal status in albino male rats are shown in Table 6. The C + MLT group showed significant increases (*p* < 0.05) in CBC parameters such as RBCs, Hb, MHCH, TLC, Nutro, Lympho, and Monocy when compared to C − MLT during MLT exposure. However, RDW decreased significantly in the C + MLT group compared to C − MLT. Furthermore, during the return to normal status after MLT exposure, the CBC parameters indicated an improvement with OEO and ThEO therapy, with no significant difference identified in RBCs, Hb, TLC, platelets, or neutrophils across groups. Furthermore, there was a significant difference in Hct, MCV, MHCH, lympho, and monocy between C − MLT and treated groups (OEO and ThEO).

### 3.3. Histopathological Examination

#### 3.3.1. Histopathology of Liver in C − MLT and C + MLT during MLT Exposure

Liver biopsies of the C − MLT group showed a consistent hepatic pattern with normal nodules and central veins, as well as polygonal hepatocytes organized into hexagonal lobules with central veins. Sinusoids contain Kupffer cells and portal triads. Lymphocyte infiltrations were found around the hepatic sinusoids (Figure 2). Liver tissue sections in the C + MLT showed disrupted liver architectures, hepatocellular necrosis, inflammation, lipid accumulation, Kupffer cell activation, altered blood flow, central vein and sinusoidal dilation, and grade three hepatic fibrosis (Figure 3).

#### 3.3.2. Histopathology of Kidney in C − MLT and C + MLT during MLT Exposure

A histological analysis in the C − MLT group of healthy kidney tissue revealed renal corpuscles, glomeruli, renal tubules, and collecting ducts. Glomeruli are round or oval structures with a central capillary tuft surrounded by the Bowman’s capsule. The renal tubules form loops of Henle in the medulla, with blood vessels, fibroblasts, and inflammatory cells in the interstitial tissue between the tubules (Figure 4). However, as shown in Figure 5, a histological analysis after MLT exposure (C + MLT) revealed that the rat’s renal tissue showed signs of glomerular hypertrophy, structural damage, degenerative changes, interstitial inflammation, and fibrosis. Extravasation and hemorrhage were also observed, indicating a chronic inflammatory process.

#### 3.3.3. Histopathology of Liver in OEO and ThEO during Return to Normal Status

Histological sections of the OEO group showed reduced hepatocellular damage, inflammation, and normal liver architecture. The oil treatment also reduced hepatic fibrosis and lipid accumulation but dilated the central vein. This suggests that OEO oil may be able to serve as a protective therapy (Figure 6). The histological sections of ThEO showed a nearly normal pattern of liver architecture, with few remaining necrosis areas, pyknotic nuclei, or inflammatory cell infiltrates. Additionally, steatosis, or the accumulation of lipid droplets within hepatocytes, was observed, indicating disrupted lipid metabolism. Mild fibrosis persisted, with dilated central veins (Figure 7).

#### 3.3.4. Histopathology of Renal in OEO and ThEO during Return to Normal Status

In this study, histological sections of renal tissue from rats exposed to MLT and treated with OEO showed reduced tubular damage, reduced interstitial inflammation, and preserved renal tissue architecture. The architecture of renal tissue appeared to be preserved, with less disruption of glomerular size and the thickness of Bowman’s capsule. However, hemorrhage was detected in some areas of the renal parenchyma (Figure 8). A histological analysis of the ThEO revealed a complex pattern of protection against Thymol’s nephrotoxic effect, as evidenced by a reduction in glomerular size and a decrease in Boymen’s capsule thickness, as well as tubular dilation, extravasation and hemorrhage, and interstitial inflammation (Figure 9).

## 4. Discussion

This study successfully examined an experimental therapeutic approach using OEO and ThEO extracts against MLT toxicity in rats. Essential oils were the most commercially successful candidate among the natural anti-toxins. This study revealed the toxic effects of disturbances in biochemical and histological factors caused by exposure to MLT. Also, the findings in the present study revealed that oral supplementation of OEO and ThEO markedly improved the return to normal status after exposure to MLT; the compounds improved liver and kidney functions and antioxidant biometric parameters. Moreover, the OEO and ThEO therapies enhanced the liver and kidney histological parameters.

Our study showed that C + MLT underwent a significant reduction of FW and BWG compared to the C − MLT group. This result is consistent with earlier research that showed morphologic and clinical changes in parameter measurements after being exposed to MLT [27,28].

The study found a significant increase in serum T. protein, Alb, Glb, and creatinine between the C − MLT and C + MLT groups following MLT exposure. However, after MLT exposure, rat renal tissue showed signs of hypertrophy, structural damage, degenerative changes, inflammation, fibrosis, extravasation, and hemorrhage, indicating chronic inflammation. While macrophages emit pro-inflammatory mediators such as interleukin-1, nitrite oxide, alpha-tumor [29], and necrosis factor in response to tissue damage, necrotic cells release pro-inflammatory mediators that worsen poison-induced liver injury [30].

Our findings also demonstrated that OEO and ThEO could reduce the lipid profile and improve antioxidant activity, while the OEO therapeutic group revealed a significant rise in serum antioxidant profile compared to C − MLT. Numerous studies have demonstrated the antioxidant qualities of ThEO and OEO, which are consistent with our findings [31,32]. Based on these data, it would seem that EO, due to its antioxidant qualities, could enhance the antioxidant activity in the treatment groups by preventing the generation of ROS [33].

The OEO oil treatment reduced hepatocellular damage, inflammation, and liver architecture alterations while reducing fibrosis and lipid accumulation, suggesting that OEO oil may be a protective therapy. Also, ThEO treatment resulted in a normal liver architecture and status and mild fibrosis grade. According to research by El-Gendy et al., OEO inhibits liver fibrosis progression through its antioxidant, anti-inflammatory, and anti-apoptotic qualities, making it a hepato-protective natural substance [34]. Our findings support the theories of Chakraborty et al., who proposed that MLT can cause liver damage and apoptosis in hepatocytes [35].

Furthermore, during the return to normal status after MLT exposure, the CBC parameters indicated an improvement with OEO and ThEO therapy. The results of this study were consistent with those of Ribeiro et al., which indicated the success of carvacrol, extracted from oregano and thyme oils as an anti-inflammatory and antioxidant compound, in improving the CBC parameters [36].

## 5. Conclusions

This study demonstrated that exposure to MLT causes toxic effects, including disruptions in biochemical and histological parameters. The results of this study also showed that oral supplementation with OEO and ThEO significantly enhanced the return to normal status following exposure to MLT; additionally, these compounds enhanced the antioxidant biometric parameters in terms of biochemical liver and kidney functions. Furthermore, the liver and kidney histological parameters were improved by the OEO and ThEO therapies.

The study’s findings allow for the presentation of a therapeutic strategy that uses OEO and ThEO to reduce MLT toxicity. Acquiring this knowledge would be the first step toward creating possible one-of-a-kind remedies with natural antitoxins. However, more pre-clinical and clinical research is required, with a particular emphasis on determining safe doses.

## Figures and Tables

**Figure 1 animals-14-02497-f001:**
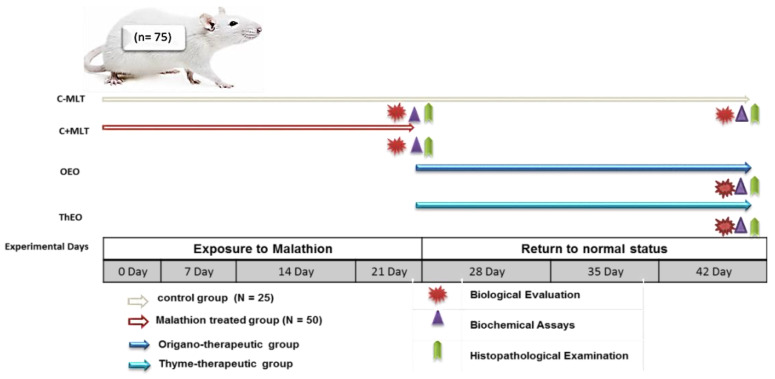
Schematic diagram showing the experimental protocol for the hazards of exposure to malathion and the efficiency of thyme and oregano essential oils as anti-toxic therapies in the rats. Rats were randomly assigned to two groups: (i) C − MLT, in which 25 rats served as the control group; and (ii) C + MLT, where 50 rats were dosed at 5 mg/kg body weight. C − MLT was continued from the earlier experiment, while MLT exposure was terminated after 21 days. The return to a normal state was determined by dividing C + MLT into two equal subgroups and treating them with 100 mg/kg body weight of OEO and ThEO, respectively.

**Figure 2 animals-14-02497-f002:**
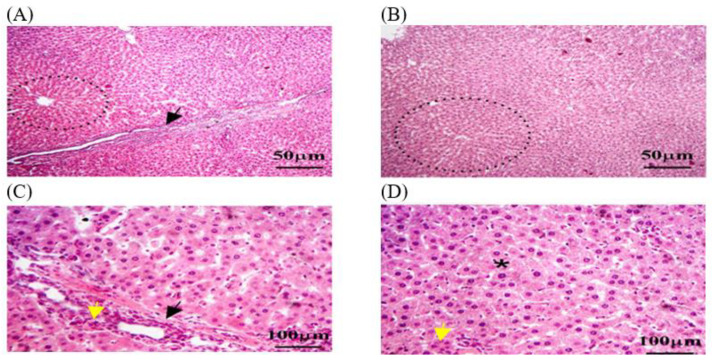
The liver tissue sections in the C − MLT group were stained with H&EC − MLT. (**A**,**B**) dotted black circles: normal hepatic nodules. (**A**,**C**) Black arrow: Sinusoids contained numerous Kupffer cells. (**C**,**D**) Yellow arrow: hepatic macrophages. (**D**) Black star: hexagonal lobules, each with a central vein at its core. The magnification power is ×100 (**A**,**B**) and ×400 (**C**,**D**).

**Figure 3 animals-14-02497-f003:**
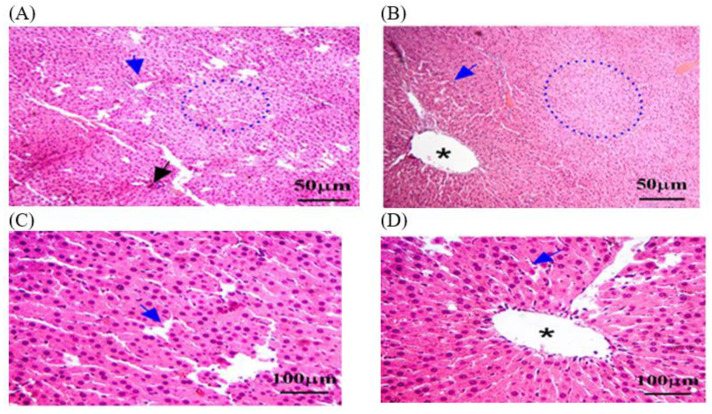
The liver tissue sections were stained with H&E in the C + MLT group. (**A**) Black arrow: vacuolar degeneration or fatty change. (**A**,**B**) dotted blue circle: liver architecture may have been disrupted, with areas of hepatocellular necrosis and inflammation. (**B**,**D**) Black star: central veins and sinusoids may be dilated. (**A**–**D**) Blue arrow: a marked degree of hepatic fibrosis. The magnification power is ×100 (**A**,**B**) and ×400 (**C**,**D**).

**Figure 4 animals-14-02497-f004:**
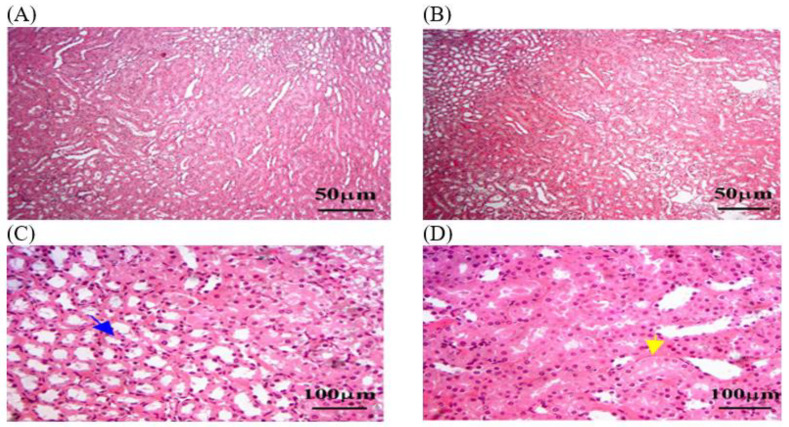
The kidney tissue sections in the C − MLT group were stained with H&EC − MLT. (**A**,**B**) Interstitial inflammation and fibrosis may have been present, reflecting a chronic inflammatory process. (**C**) Blue arrow: Bowman’s capsules. (**D**) Yellow arrow: degenerative changes, such as vacuolization or necrosis. The magnification power is ×100 (**A**,**B**) and ×400 (**C**,**D**).

**Figure 5 animals-14-02497-f005:**
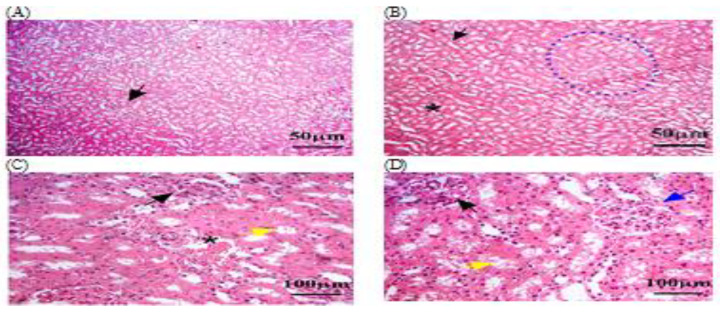
The kidney tissue sections were stained with H&E in the C + MLT group. (**A**–**D**) Black arrow: glomerular hypertrophy and hypercellularity. (**D**) Blue arrow: Bowman’s capsules. (**C**,**D**) Yellow arrow: degenerative changes, such as vacuolization or necrosis. (**B**,**C**) Black star: extravasation and hemorrhage; dotted blue circle: appeared disrupted, with areas of hepatocellular necrosis. The magnification power is ×100 (**A**,**B**) and ×400 (**C**,**D**).

**Figure 6 animals-14-02497-f006:**
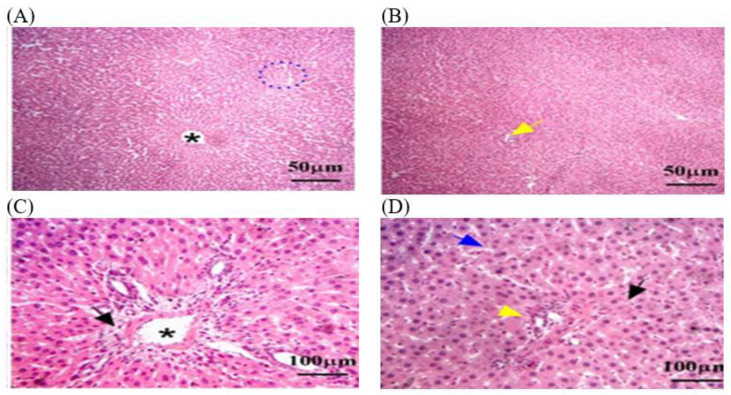
The liver tissue sections stained with H&E in the OEO group returned to normal status after MLT exposure. (**A**) Blue dotted circle: necrosis and psychotic nuclei compared to rats exposed to MLT alone. (**B**,**D**) yellow arrow: inflammation parenchyma, with fewer infiltrates of inflammatory cells such as lymphocytes and macrophages. (**D**) Blue arrow: a moderate reduction in hepatic fibrosis. (**C**,**D**) Black arrow: a few hepatocytes with vacuolated cytoplasm, indicating lipid accumulation. (**A**,**C**) Black star: dilated central vein was detected. The magnification power is ×100 (**A**,**B**) and ×400 (**C**,**D**).

**Figure 7 animals-14-02497-f007:**
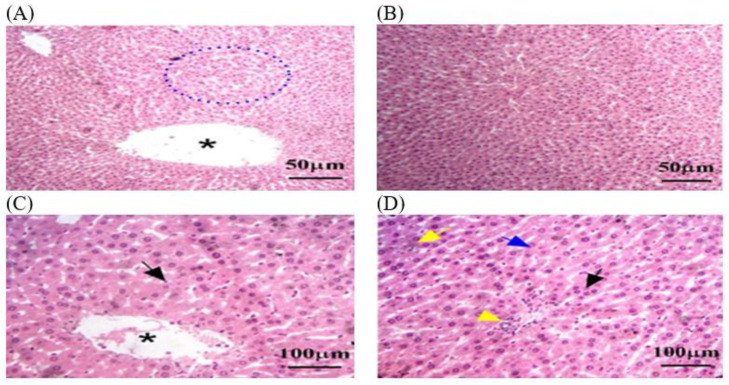
The liver tissue sections stained with H&E in the ThEO group returned to normal status after MLT exposure. (**A**) Blue dotted circle: few remaining necrosis areas, psychotic nuclei, and inflammatory cell infiltrates. (**D**) Yellow arrow: accumulation of lipid droplets within hepatocytes. (**D**) Blue arrow: fibrosis grade persists. (**A**,**C**) Black star: dilated central veins. (**C**,**D**) Black arrow: hepatocellular necrosis by the presence of pyknotic nuclei and disrupted hepatic cords. The magnification power is ×100 (**A**,**B**) and ×400 (**C**,**D**).

**Figure 8 animals-14-02497-f008:**
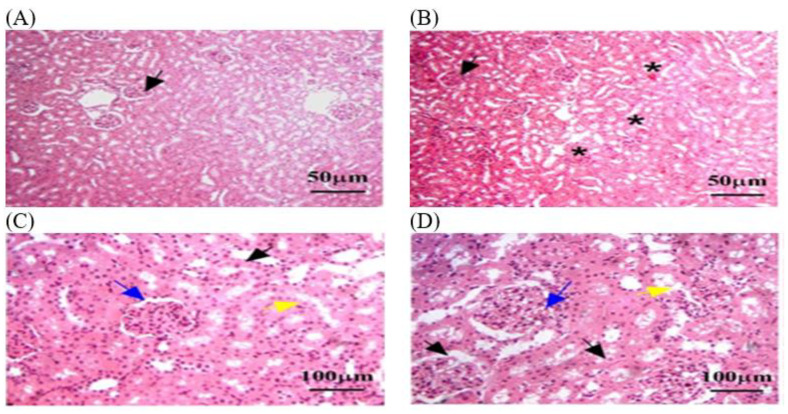
The kidney tissue sections stained with H&E in the OEO group returned to normal status after MLT exposure. (**A**–**D**) Black arrow: a nearly normal pattern of liver architecture, with few remaining necrosis areas. (**B**) Black star: dilated central veins. (**C**,**D**) Blue arrow: Mild fibrosis grade persists. (**C**,**D**) Yellow arrow: steatosis, or the accumulation of lipid droplets within hepatocytes. The magnification power is ×100 (**A**,**B**) and ×400 (**C**,**D**).

**Figure 9 animals-14-02497-f009:**
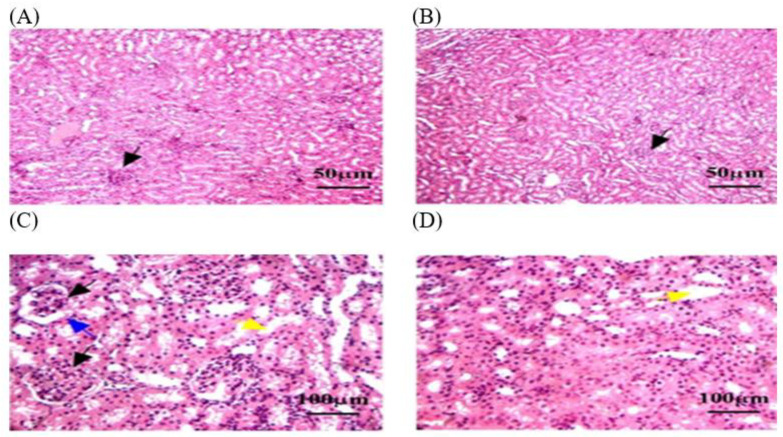
The kidney tissue sections stained with H&E in the ThEO group return to normal status after MLT exposure. (**A**–**C**) Black arrow: a reduction in glomerular size. (**C**) Blue arrow: a decrease in Boymen’s capsule thickness. (**C**,**D**) Yellow arrow: tubular dilation. The magnification power is ×100 (**A**,**B**) and ×400 (**C**,**D**).

**Table 1 animals-14-02497-t001:** Certificate of ThEO and OEO analysis *.

Items	Analysis	
	ThEO	OEO
Batch No.	TO/CAL/5021/21-22	
Country of origin	India	
Appearance	Yellow to pale yellow liquid	Yellow to amber-colored liquid
Odor	The characteristic odor of thyme and sharp burning taste	pungent, spicy aroma
Solubility in water	Insoluble	Insoluble
Specific gravity	0.919 (0.900–0.930)	0.9370–0.9380
Refraction index at 25 °C	1.4998 (1.4900–1.5100)	1.510–1.520
Content	50.21% (50.00% minimum)	50.85% (50.00% minimum)

* performed by Monachem, a strategic partner for specialty chemicals and a certified contract manufacturer in India.

**Table 2 animals-14-02497-t002:** The initial weight (IW; g.), Final weight (FW; g.), and body weight gain (BWG; g.) in the C − MLT and C + MLT during exposure to malathion and C − MLT, OEO, and ThEO during return to normal status in the albino male rats.

**Item**		**C − MLT**	**C + MLT**		***p*-Value**
Exposure to Malathion (21 day)					
IW	at 0 day	135.10 ± 1.06 ^ns^	134.80 ± 0.59 ^ns^		0.808
FW	at 21 day	192.00 ± 3.35 **	167.50 ± 4.84		0.001
BWG		56.90 ± 3.94 **	32.70 ± 4.71		0.001
Return to normal status (42 day)					
Item		C − MLT	OEO	ThEO	*p*-Value
IW	at 21 day	190.00 ± 2.89 ^a^	163.33 ± 1.20 ^b^	163.67 ± 2.73 ^b^	0.001
FW	at 42 day	216.33 ± 2.60 ^a^	187.67 ± 1.86 ^b^	191.67 ± 2.91 ^b^	0.001
BWG		26.33 ± 0.33 ^a^	24.33 ± 0.67 ^b^	28.00 ± 0.58 ^a^	0.009

IW = initial weight, FW = Final weight, BWG = body weight gain, C − MLT = control group, C + MLT = malathion group, OEO = oregano essential oil group, ThEO = thyme essential oil group. ^ns^ = Significance at *p* > 0.05. ** = Significance at *p* < 0.01, ^a,b^ = Duncan test.

**Table 3 animals-14-02497-t003:** Liver functions (U/L; ALT, AST, ALK-ph, and AChE) in the C − MLT and C + MLT during exposure to malathion and C − MLT, OEO, and ThEO during return to normal status in the albino male rats.

**Item**	**C − MLT**	**C + MLT**		***p*-Value**
Exposure to Malathion (21 day)				
ALT	46.40 ± 0.22	73.20 ± 0.66 **		0.001
AST	153.10 ± 1.88	184.90 ± 0.33 **		0.001
ALK-Ph	109.70 ± 1.22	193.20 ± 2.65 **		0.001
AChE	5.75 ± 0.09	1.28 ± 0.03 **		0.001
Return to normal status (42 day)				
Item	C − MLT	OEO	ThEO	*p*-Value
ALT	46.50 ± 0.29 ^a^	28.50 ± 2.02 ^c^	42.00 ± 0.58 ^b^	0.001
AST	152.25 ± 2.45 ^a^	149.50 ± 0.29 ^a^	114.50 ± 0.29 ^b^	0.001
ALK-Ph	110.25 ± 1.59 ^a^	112.00 ± 1.73 ^a^	103.00 ± 1.15 ^b^	0.013
AChE	5.70 ± 0.10 ^a^	4.88 ± 0.02 ^b^	4.38 ± 0.25 ^c^	0.001

C − MLT = control group, C + MLT = malathion group, OEO = oregano essential oil group, ThEO = thyme essential oil group. ** = Significance at *p* < 0.01, ^a,b,c^ = Duncan test.

**Table 4 animals-14-02497-t004:** Kidney functions enzymes and lipid profiles in the C − MLT and C + MLT during exposure to malathion and C − MLT, OEO, and ThEO during return to normal status in the albino male rats.

**Item**	**C − MLT**	**C + MLT**		***p*-Value**
Exposure to Malathion (21 day)				
Glucose (mg/dL)	81.33 ± 1.86	177.00 ± 10.44 *		0.010
T. protein (g/dL)	6.89 ± 0.29 *	5.10 ± 0.04		0.023
Alb (g/dL)	5.23 ± 0.17 **	1.79 ± 0.41		0.006
Glb (g/dL)	3.17 ± 0.04 *	2.11 ± 0.15		0.014
Creatinine (mg/dL)	0.57 ± 0.02 **	0.95 ± 0.01		0.001
TC (mg/dL)	81.03 ± 0.77 *	60.50 ± 2.93		0.015
TG (mg/dL)	60.74 ± 0.52 ^ns^	50.54 ± 4.91 ^ns^		0.172
VLDL (g/dL)	13.96 ± 0.98 ^ns^	12.59 ± 0.82 ^ns^		0.346
HDL (g/dL)	61.12 ± 0.88 **	38.30 ± 1.18		0.001
LDL (g/dL)	9.13 ± 0.39 **	6.42 ± 0.38		0.008
AI	0.47 ± 0.06	0.94 ± 0.05 **		0.003
Return to normal status (42 day)				
Item	C − MLT	OEO	ThEO	*p*-Value
Glucose (mg/dL)	89.67 ± 2.91 ^b^	145.00 ± 1.73 ^a^	151.67 ± 1.45 ^a^	0.001
T. protein (g/dL)	7.22 ± 0.16 ^ns^	5.76 ± 0.57 ^ns^	6.49 ± 0.66	0.214
Alb (g/dL)	4.86 ± 0.26 ^a^	2.84 ± 0.14 ^b^	2.79 ± 0.17 ^b^	0.001
Glb (g/dL)	3.43 ± 0.23 ^a^	1.60 ± 0.22 ^b^	1.95 ± 0.04 ^b^	0.001
Creatinine (mg/dL)	0.59 ± 0.01 ^ns^	0.68 ± 0.10 ^ns^	0.71 ± 0.13	0.653
TC (mg/dL)	80.93 ± 2.34 ^a^	64.19 ± 2.32 ^b^	64.22 ± 2.19 ^b^	0.003
TG (mg/dL)	63.34 ± 1.30 ^a^	54.61 ± 2.15 ^b^	57.53 ± 0.57 ^b^	0.016
VLDL (g/dL)	13.75 ± 0.15 ^ns^	11.78 ± 0.27 ^ns^	11.63 ± 0.62	0.120
HDL (g/dL)	61.37 ± 0.84 ^a^	42.34 ± 1.60 ^b^	45.82 ± 2.03 ^b^	0.001
LDL (g/dL)	9.32 ± 0.34 ^a^	7.40 ± 0.28 ^b^	7.82 ± 0.50 ^b^	0.028
AI	0.38 ± 0.02 ^ns^	0.51 ± 0.03 ^ns^	0.42 ± 0.06	0.134

C − MLT = control group, C + MLT = malathion group, OEO = oregano essential oil group, ThEO = thyme essential oil group. ^ns^ = Significance at *p* > 0.05. * = Significance at *p* < 0.05, ** = Significance at *p* < 0.01, ^a,b^ = Duncan test.

**Table 5 animals-14-02497-t005:** The antioxidant profile (GPX, BCA, TAC, SOD, and MDA) in the C − MLT and C + MLT groups during exposure to malathion and C − MLT, OEO, and ThEO during return to normal status in albino male rats.

**Item**	**C − MLT**	**C + MLT**		***p*-Value**
Exposure to Malathion (21 day)				
TAC	7.30 ± 0.13 **	2.21 ± 0.01		0.001
GPX	4.41 ± 0.21 **	3.00 ± 0.09		0.001
SOD	14.55 ± 0.28 **	17.38 ± 0.29		0.001
BCA	111.19 ± 4.23	179.40 ± 2.43 **		0.001
MDA	29.06 ± 0.53	40.12 ± 0.37 **		0.001
Return to normal status (42 day)				
Item	C − MLT	OEO	ThEO	*p*-Value
TAC	7.24 ± 0.17 ^b^	13.33 ± 0.79 ^a^	7.97 ± 0.13 ^b^	0.001
GPX	4.27 ± 0.36 ^a^	7.31 ± 0.47 ^b^	16.66 ± 0.64 ^c^	0.001
SOD	11.15 ± 0.38 ^b^	14.68 ± 0.36 ^a^	11.78±0.30 ^b^	0.001
BCA	109.28 ± 5.53 ^ns^	120.00 ± 0.58 ^ns^	106.00 ± 1.15 ^ns^	0.112
MDA	29.30 ± 0.69 ^b^	37.00 ± 1.44 ^a^	32.150.55 ^b^	0.002

C − MLT = control group, C + MLT = malathion group, OEO = oregano essential oil group, ThEO = thyme essential oil group. ^ns^ = Significance at *p* > 0.05. ** = Significance at *p* < 0.01, ^a,b,c^ = Duncan test.

**Table 6 animals-14-02497-t006:** CBCs in the C − MLT and C + MLT during exposure to malathion and C − MLT, OEO, and ThEO during return to normal status in the albino male rats.

**Item**	**C − MLT**	**C + MLT**		***p*-Value**
Exposure to Malathion (21 day)				
RBCs	6.69 ± 0.02	8.40 ± 0.20 *		0.013
Hb	14.67 ± 0.04	16.85 ± 0.49 *		0.046
Hct	44.80 ± 0.03 ^ns^	44.90 ± 0.06 ^ns^		0.221
MCV	66.66 ± 0.12 **	53.54 ± 1.22		0.008
MCH	21.66 ± 0.06 **	20.10 ± 0.13		0.002
MHCH	32.55 ± 0.08	37.52 ± 1.05 *		0.041
RDW	24.52 ± 0.20 **	15.10 ± 0.23		0.001
Platelets	739.67 ± 10.11 ^ns^	844.00 ± 49.65 ^ns^		0.166
TLC	11.42 ± 1.39	21.29 ± 0.20 *		0.018
Nutro	8.50 ± 2.60	20.67 ± 0.88 *		0.032
Lympho	73.00 ± 1.15	83.00 ± 2.31 *		0.032
Monocy	6.67 ± 0.33	8.50 ± 0.29 *		0.015
Return to normal status (21 day)				
Item	C − MLT	OEO	ThEO	*p*-Value
RBCs	6.50 ± 0.15 ^ns^	6.21 ± 0.05 ^ns^	6.35 ± 0.21 ^ns^	0.441
Hb	14.43 ± 0.12 ^ns^	14.50 ± 0.12 ^ns^	14.75 ± 1.01 ^ns^	0.924
Hct	43.10 ± 0.70 ^a^	34.85 ± 0.61 ^c^	39.50 ± 0.87 ^b^	0.001
MCV	66.09 ± 0.31 ^a^	56.21 ± 1.48 ^c^	62.27 ± 0.68 ^b^	0.001
MCH	21.35 ± 0.17 ^b^	23.38 ± 0.39 ^c^	23.15 ± 0.83 ^bc^	0.072
MHCH	31.19 ± 0.46 ^c^	41.63 ± 0.39 ^b^	37.32 ± 1.79 ^a^	0.002
RDW	23.73 ± 0.62 ^a^	15.80 ± 0.12 ^b^	23.25 ± 2.68 ^a^	0.021
Platelets	753.33 ± 4.18 ^ns^	843.00 ± 3.46 ^ns^	809.50 ± 59.76 ^ns^	0.258
TLC	16.64 ± 1.40 ^ns^	19.17 ± 0.53 ^ns^	21.02 ± 1.80 ^ns^	0.150
Nutro	12.33 ± 1.45 ^ns^	6.50 ± 0.87 ^ns^	11.00 ± 2.89 ^ns^	0.162
Lympho	74.50 ± 1.22 ^b^	86.00 ± 1.15 ^a^	80.00 ± 2.89 ^a^	0.016
Monocy	6.00 ± 0.58 ^c^	7.50 ± 0.29 ^b^	9.00 ± 0.00 ^a^	0.004

C − MLT = control group, C + MLT = malathion group, OEO = oregano essential oil group, ThEO = thyme essential oil group. ^ns^ = Significance at *p* > 0.05. * = Significance at *p* < 0.05, ** = Significance at *p* < 0.01, ^a,b,c^ = Duncan test.

## Data Availability

The raw data supporting the conclusions of this article will be made available by the authors without undue reservation.

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
