# Peer review of "Efficiency of Thyme and Oregano Essential Oils in Counteracting the Hazardous Effects of Malathion in Rats"

_animals, 2024, doi:10.3390/ani14172497_

Round 1

Reviewer 1 Report

Comments and Suggestions for Authors

Please find comments on the attached pdf file.

Serious ethics concern: why would you use rats when you could use non-mammalian models, e.g., insects? You know that rats are mammalians, just like us humans. As such, they are intelligent and emotional. Your study group basically poisoned and "euthanized" intelligent and emotional beings. This is utterly unethical and would be unacceptable in my lab.  

Comments on the Quality of English Language

Please explain acronyms and initialisms throughout the whole paper, check English for minor inconsistencies and correct scattered typos.

Author Response

Dear Reviewer 1

We appreciate you for your precious time in reviewing our manuscript. We have carefully considered your comments and tried our best to address them to make an extensive revision. We hope the manuscript's careful revisions meet your high standards. Our detailed, point-by-point responses to the comments are given below.

  • Journal: Animals(ISSN 2076-2615).
  • Manuscript ID: animals-3101075.
  • Type: Article.
  • Title: Efficiency of thyme and oregano essential oils on the hazardous effects of Melathion in rats

#- Comments and Suggestions for Authors

Please find comments on the attached pdf file "peer-review-38545254.v1.pdf".

Comments 1: [In the title. Correct the word melathion].

Response 1: [Thank you very much for your comment, it has been corrected].

Comments 2: [Please clarify acronyms and initialisms throughout the manuscript, e.g., body weight (BW(].

Response 2: [Thank you very much for your comment. The manuscript has been reviewed in general and the missing acronyms and initialisms have been clarifyed.].

Comments 3: [Please list keywords in alphabetical order].

Response 3: [Thank you very much for your comment. The list keywords were updated as alphabetical].

Comments 4: [Please add 1-3 sentences to explain the mechanism of action of MLT: how does it work on pests?].

Response 4: [Thank you very much for your comment. The sentences were added in the line from 52 to 55 "MLT's toxicity is due to its mode of action as an acetylcholinesterase (AChE) inhibitor. The inhibition of AChE causes acetylcholine to accumulate at muscarinic and nicotinic sites, resulting in acute hyperstimulation due to the neurotransmitter's persistent presence [3]"].

Comments 5: [Correct the word melathion].

Response 5: [Thank you very much for your comment, it has been corrected].

Comments 6: [I surmise that significance was considered for p < 0.05 It wouldn't hurt to clarify this in the text].

Response 6: [Thank you very much for your comment, the "The values ≤ 0.05 of significance were considered" was added in the line 155.].

Comments 7: [Looks like it's actually C+MLT].

Response 7: [Thank you very much for your correct comment, please accept our apologies for this error "has been corrected"].

Comments 8: [Isn't it C+MLT?].

Response 8: [Thank you very much for your comment. All figures titles have been reviewed and corrected].

Comments 9: [add "in rats" in the line 298].

Response 9: [Thank you very much; "in rats" has been added].

Comments 10: [remove "toxicity" in the line 301.].

Response 10: [Thank you very much; "toxicity" was removed].

Comments 11: [Serious ethics concern: why would you use rats when you could use non-mammalian models, e.g., insects? You know that rats are mammalians, just like us humans. As such, they are intelligent and emotional. Your study group basically poisoned and "euthanized" intelligent and emotional beings. This is utterly unethical and would be unacceptable in my lab. ].

Response 11: [Thank you very much for your valuable comment. We completely agree with you that mice are animals with feelings and emotions, and we value that fact. On the other hand, rats have contributed the lifeblood of all living species, including humans, animals, and plants, by offering a wide range of therapeutic services and vaccines that provide radical remedies to a variety of issues and diseases.

While Mice and rats make up approximately 95% of all laboratory animals, with mice the most commonly used animal in biomedical research. Mice are a commonly selected animal model. Mice have been used as research subjects for studies ranging from biology to psychology to engineering. They are used to model human diseases for the purpose of finding treatments or cures. Some of the diseases they model include: hypertension, diabetes, cataracts, obesity, seizures, respiratory problems, deafness, Parkinson's disease, Alzheimer's disease, various cancers, cystic fibrosis, and acquired immunodeficiency syndrome (AIDS), heart disease, muscular dystrophy, and spinal cord injuries. Mice are also used in behavioral, sensory, aging, nutrition, and genetic studies. This list is in no way complete as geneticists, biologists, and other scientists are rapidly finding new uses for the domestic mouse in research.

Furthermore, all Institutional and National Guidelines for the care and use of animals were followed in accordance with the Egyptian Medical Research Ethics Committee (no. 14-126), and the Research Ethics Committee of the Faculty of Agriculture at Assiut University granted ethical approval for the aforementioned research project (Reference No: 03-2024-0006.

Comments 12: [Comments on the Quality of English Language. Please explain acronyms and initialisms throughout the whole paper, check English for minor inconsistencies and correct scattered typos.].

Response 12: [We have employed an English-speaking colleague to achieve a comprehensive linguistic review that we hope will meet your valuable requirements.]

We would like to thank you once again for your consideration of our work and inviting us to submit the revised manuscript.

Reviewer 2 Report

Comments and Suggestions for Authors

The manuscript titled "Efficiency of thyme and oregano essential oils on the hazardous effects of Melathion in rats" By Al-Saeed et al submitted to Animals for consideration aims to affirm the antitoxic claims of OEO and ThEO against MLT toxicity. Also, to investigate the amount of anti-toxic capacity on biochemical and histological effects. 

The following comments are needed to be considered:

- Information given in the Abstract regarding experimental design is unclear and needs to be rephrased (Lines 19-23).

- At the end of the abstract (Lines ), probably you mean natural toxins ... not antitoxins.

- In the introduction, the first part does not give a logical idea about malathion, as it sometimes mentions the role of malathion and sometimes mentions its harmful effects in the same sentence. Please divide it more comprehensively so that you give an idea about malathion and its uses, then move to the possibility of it reaching mammals, and then move to its toxic effects on mammals.

- At the end of the introduction, you mentioned: "investigate the amount of anti-toxic capacity". You used one dose from each EO. How can you investigate the amount of anti-toxic capacity with one dose? It is not possible as you have to use different doses to investigate which one will work better as an anti-toxic.

- Experimental design lacks information regarding the insecticide used (formulation type, concentration ... etc). The concentration/purity of 99.9% is not correct. The number of animals is very high for this experiment. The groups and names are confusing and not understandable. Why did you use EOs after exposure and not during exposure?

- The method of administration and time are not mentioned.

Comments on the Quality of English Language

The language needs extensive revision.

Author Response

Dear Reviewer 2

We appreciate you for your precious time in reviewing our manuscript. We have carefully considered your comments and tried our best to address them to make an extensive revision. We hope the manuscript's careful revisions meet your high standards. Our detailed, point-by-point responses to the comments are given below.

  • Journal: Animals(ISSN 2076-2615).
  • Manuscript ID: animals-3101075.
  • Type: Article.
  • Title: Efficiency of thyme and oregano essential oils on the hazardous effects of Melathion in rats

#- Comments and Suggestions for Authors

Comments 1: [The manuscript titled "Efficiency of thyme and oregano essential oils on the hazardous effects of Melathion in rats" By Al-Saeed et al submitted to Animals for consideration aims to affirm the antitoxic claims of OEO and ThEO against MLT toxicity. Also, to investigate the amount of anti-toxic capacity on biochemical and histological effects].

Response 1: [We appreciate you for your precious time in reviewing our manuscript. We have carefully considered your comments and tried my best to address them to make an extensive revision. We hope the manuscript's careful revisions meet your high standards. Our detailed, point-by-point responses to the comments are given below].

Comments 2: [Information given in the Abstract regarding experimental design is unclear and needs to be rephrased (Lines 19-23)].

Response 2: [Thank you very much for your comment. We have updated experimental design in the Abstract section and we hope that we have met your valuable requirements (Lines 25-31)].

Comments 3: [At the end of the abstract (Lines), probably you mean natural toxins ... not antitoxins.].

Response 3: [Thank you very much for your comment; yes we mean natural toxins ... not antitoxins. The modification has been made in line no 44]

Comments 4: [- In the introduction, the first part does not give a logical idea about malathion, as it sometimes mentions the role of malathion and sometimes mentions its harmful effects in the same sentence. Please divide it more comprehensively so that you give an idea about malathion and its uses, then move to the possibility of it reaching mammals, and then move to its toxic effects on mammals.]

Response 4: [Thank you very much for your suggestion to rearrange and write the initial part of the introduction section. This section has been updated as follows:

  • Lines 49-51 show an idea about MLT.
  • Lines 52-55 show the mode of MLT action.
  • Lines 56-58 show MLT uses.
  • Lines 58-61 show MLT possibility of it reaching mammals.
  • Lines 61-64 show MLT toxic effects on mammals.

Comments 5: [At the end of the introduction, you mentioned: "investigate the amount of anti-toxic capacity". You used one dose from each EO. How can you investigate the amount of anti-toxic capacity with one dose? It is not possible as you have to use different doses to investigate which one will work better as an anti-toxic.].

Response 5: [Thank you very much for your valuable comment. The sentence at the end of the introduction has been updated to (Lines 80-83)“As a result, this study was carried out to investigate the biochemical and histological hazardous reactions to MLT and the efficiency of ThEO and OEO essential oils as anti-toxic therapies to return to a natural state after MLT exposure. Such knowledge may be a step toward developing potentially unique treatment options for natural antitoxins.”and at the end of the extract, we referred to “ However, more pre-clinical and clinical research is required, with a particular emphasis on determining safe doses"].

Comments 6: [Experimental design lacks information regarding the insecticide used (formulation type, concentration ... etc). The concentration/purity of 99.9% is not correct.

Response 6: [Thank you very much for your thoughtful comment. The information required after requesting the Safety Data Sheet has been provided by the company from which the melathion is obtained and we are prepared to meet the value requirements.].

Comments 7: [The number of animals is very high for this experiment.].

Response 7: [Thank you very much. This number was used to later divide the group into two subgroups treated with essential oils].

Comments 8: [The groups and names are confusing and not understandable.].

Response 8: [Thank you very much for your valuable comment. We have tried to use group names as follows:

  • C-MLT is the control group
  • C+MLT is the group exposed to MLT
  • OEO Oregano Oil Treatment Group
  • ThEO group treated with thymol oil

We are ready to accept, rename and formulate any proposal adopted to benefit the study and be easier for the reader.

Comments 9: [Why did you use EOs after exposure and not during exposure?].

Response 9: [Thank you very much for your thoughtful comment. The aim of the study was the efficacy of essential oils to return to the normal state after toxicity "this study was carried out to investigate the biochemical and histological hazardous reactions to MLT and the efficiency of ThEO and OEO essential oils as anti-toxic therapies to return to a natural state after MLT exposure.", We are looking forward to a new study in our laboratory in which natural oils are used during exposure to toxicity].

Comments 10: [The method of administration and time are not mentioned]

Response 10: [Thank you very much,

  • we have provided animal control with lines 88-90 “The rats were kept in stainless steel cages, and housed under standard conditions of temperature (23 ± 2°C) and lighting (12 hours; light/dark cycles), with free "access to food and drinking water ad libitum."
  • The melathion exposure time is provided in line 94 “(ii) C+MLT, 50 rats dosed at 5 mg/kg/BW MLT (volume: 1 ml; MLT LD50: 1/316) for 21 days.”
  • And saving time for treatment with essential oils to return to the normal state in lines 101: “The return to a normal state was examined by dividing C+MLT into two equal subgroups (OEO and TheEO) and treating each with 100 mg/kg/BW for 21 days.”

Comments 11: [The language needs extensive revision]

Response 11: [We have employed an English-speaking colleague to achieve a comprehensive linguistic review that we hope will meet your valuable requirements.]

We would like to thank you once again for your consideration of our work and inviting us to submit the revised manuscript.

-.
